# Evolve Smoothly, Fit Consistently: Learning Smooth Latent Dynamics For Advection-Dominated Systems

**Zhong Yi Wan**\*
Google Research
Mountain View, CA 94043, USA
`wanzy@google.com`

**Leonardo Zepeda-Núñez**\*
Google Research
Mountain View, CA 94043, USA
`lzepedanunez@google.com`

**Anudhyan Boral**
Google Research
Mountain View, CA 94043, USA
`anudhyan@google.com`

**Fei Sha**
Google Research
Mountain View, CA 94043, USA
`fsha@google.com`

## Abstract

We present a data-driven, space-time continuous framework to learn surrogate models for complex physical systems described by advection-dominated partial differential equations. Those systems have slow-decaying Kolmogorov $n$-width that hinders standard methods, including reduced order modeling, from producing high-fidelity simulations at low cost. In this work, we construct hypernetwork-based latent dynamical models directly on the parameter space of a compact representation network. We leverage the expressive power of the network and a specially designed consistency-inducing regularization to obtain latent trajectories that are both low-dimensional and smooth. These properties render our surrogate models highly efficient at inference time. We show the efficacy of our framework by learning models that generate accurate multi-step rollout predictions at much faster inference speed compared to competitors, for several challenging examples.

## 1 Introduction

High-fidelity numerical simulation of physical systems modeled by time-dependent partial differential equations (PDEs) has been at the center of many technological advances in the last century. However, for engineering applications such as design, control, optimization, data assimilation, and uncertainty quantification, which require repeated model evaluation over a potentially large number of parameters, or initial conditions, high-fidelity simulations remain prohibitively expensive, even with state-of-art PDE solvers. The necessity of reducing the overall cost for such downstream applications has led to the development of surrogate models, which captures the core behavior of the target system but at a fraction of the cost.

One of the most popular frameworks in the last decades (Aubry et al., 1988) to build such surrogates has been reduced order models (ROMs). In a nutshell, they construct lower-dimensional representations and their corresponding reduced dynamics that capture the system's behavior of interest. The computational gains then stem from the evolution of a lower-dimensional latent representation (see Benner et al. (2015) for a comprehensive review). However, classical ROM techniques often prove inadequate for advection-dominated systems, whose trajectories do not admit fast decaying Kolmogorov $n$-width (Pinkus, 2012), i.e. there does not exist a well-approximating $n$-dimensional subspace with low $n$. This hinders projection-based ROM approaches from simultaneously achieving high accuracy and efficiency (Peherstorfer, 2020).

Furthermore, many classical ROM methods require exact and complete knowledge of the underlying PDEs. However, that requirement is unrealistic in most real-world applications. For example, in

---

\*Equal contribution.

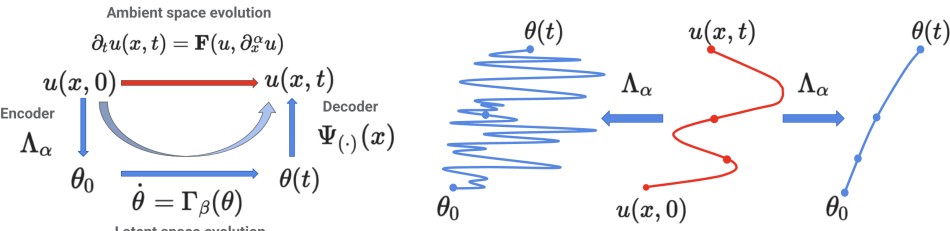

Figure 1: (left) Diagram of the latent space evolution, where $u$ represents the state variables, and $\theta$ the latent space variables, which in this case correspond to the weights of a neural network. (right) Sketch of two possible latent space trajectories, we seek to implicitly regularize the encode-decoder to obtain trajectories as the second one, which allows us to take very long time-steps in latent space.

weather and climate modeling, many physical processes are unknown or unresolved and thus are approximated. In other cases, the form of the PDE is unknown. Hence, being able to learn ROMs directly from data is highly desirable. In particular, modern learning techniques offer the opportunities of using "black-box" neural networks to parameterize the projections between the original space and the low-dimensional space as well as the dynamics in the low-dimensional space (Sanchez-Gonzalez et al., 2020; Chen et al., 2021; Kochkov et al., 2020).

On the other end, neural parameterization can be too flexible such that they are not necessarily grounded in physics. Thus, they do not extrapolate well to unseen conditions, or in the case of modeling dynamical systems, they do not evolve very far from their initial point before the trajectories diverge in a nonphysical fashion. *How can we inject into neural models the right types and amount of physical constraints?*

In this paper, we explore a hypernetwork-type surrogate model, leveraging contemporary machine learning techniques but constrained with physics-informed priors: (a) *Time-Space Continuity*: the trajectories of the model are continuous in time and space; (b) *Causality*: the present state of the model depends explicitly in the state of the model in the past; (c) *Mesh-Agnostic*: the trajectories of the model should be independent of the discretization both in space and time, so it can be sampled at any given spacial-time point (d) *Latent-Space Smoothness* if the trajectories of the model evolve smoothly, the same should be true for the latent-space trajectories. The first three properties are enforced explicitly by choices of the architectures, whereas the last one is enforced implicitly at training via a consistency regularization.

Concretely, we leverage the expressive power of neural networks (Hornik et al., 1990) to represent the state of the systems through an ansatz (decoder) network that takes cues from pseudo-spectral methods (Boyd, 2001), and Neural Radiance Fields (Mildenhall et al., 2020), whose weights encode the latent space, and are computed by an encoder-hypernetwork. This hypernetwork encoder-decoder combination is trained on trajectories of the system, which together with a consistency regularization provides smooth latent-space trajectories. The smoothness of the trajectories renders the learning of the latent-space dynamics effortless using a neural ODE (Chen et al., 2018) model whose dynamics is governed by a multi-scale network similar to a U-Net (Ronneberger et al., 2015) that extracts weight, layer and graph level features important for computing the system dynamics directly. This allows us to evolve the system *completely* in latent-space, only decoding to ambient space when required by the downstream application (such process is depicted in Fig. 1 left). In addition, due to the smooth nature of the latent trajectories, we can take very large time-steps when evolving the system, thus providing remarkable computational gains, particularly compared to competing methods that do not regularize the smoothness of the latent space (see Fig. 1 right for an illustration).

The proposed framework is nonlinear and applicable to a wide range of systems, although in this paper we specifically focus on advection-dominated systems to demonstrate its effectiveness. The rest of the paper is organized as follows. We briefly review existing work in solving PDEs in §2. We describe our methodology in §3, focusing on how to use consistency loss to learn smooth latent dynamics. We contrast the proposed methodology to existing methods on several benchmark systems in §4 and conclude in §5.

## 2 RELATED WORK

The relevant literature is extensive and we divide it into six domains, ranging from PDE solvers for known equations, to purely data-driven learned surrogates, spanning a wide spectrum of approaches.

**Fast PDE solvers** typically aim at leveraging the analytical properties of the underlying PDE to obtain low-complexity and highly parallel algorithms. Despite impressive progress (Martinsson, 2019), they are still limited by the need to mesh the ambient space, which plays an important role in the accuracy of the solution, as well as the stringent conditions for stable time-stepping.

**Classical ROM methods** seek to identify low-dimensional linear approximation spaces tailored to representing the snapshots of specific problems (in contrast with general spaces such as those in finite element or spectral methods). Approximation spaces are usually derived from data samples (Aubry et al., 1988; Chinesta et al., 2011; Barrault et al., 2004; Amsallem et al., 2012), and reduced order dynamical models are obtained by projecting the system equations onto the approximation space (Galerkin, 1915). These approaches rely on exact knowledge of the underlying PDEs and are linear in nature, although various methods have been devised to address nonlinearity in the systems (Willcox, 2006; Astrid et al., 2008; Chaturantabut & Sorensen, 2010; Geelen et al., 2022; Ayed et al., 2019).

**Hybrid Physics-ML** methods hybridize classical numerical methods with contemporary data-driven deep learning techniques (Mishra, 2018; Bar-Sinai et al., 2019; Kochkov et al., 2021; List et al., 2022; Bruno et al., 2021; Frezat et al., 2022; Dresdner et al., 2022). These approaches *learn* corrections to numerical schemes from high-resolution simulation data, resulting in fast, low-resolution methods with an accuracy comparable to the ones obtained from more expensive simulations.

**Operator Learning** seeks to learn the differential operator directly by mimicking the analytical properties of its class, such as pseudo-differential (Hörmander, 2007) or Fourier integral operators (Hörmander, 2009), but without explicit PDE-informed components. These methods often leverage the Fourier transform (Li et al., 2021; Tran et al., 2021), the off-diagonal low-rank structure of the associated Green's function (Fan et al., 2019; Li et al., 2020a), or approximation-theoretic structures (Lu et al., 2021).

**Neural Ansatz** methods aim to leverage the approximation properties of neural networks (Hornik et al., 1990), by replacing the usual linear combination of handcrafted basis functions by a more general neural network ansatz. The physics prior is incorporated explicitly by enforcing the underlying PDE in strong (Raissi et al., 2019; Eivazi et al., 2021), weak (E & Yu, 2018; Gao et al., 2022), or min-max form (Zang et al., 2020). Besides a few exceptions, e.g., (de Avila Belbute-Peres et al., 2021; Bruna et al., 2022), these formulations often require solving highly non-convex optimization problems at inference time.

**Purely Learned Surrogates** fully replace numerical schemes with surrogate models learned from data. A number of different architectures have been explored, including multi-scale convolutional neural networks (Ronneberger et al., 2015; Wang et al., 2020), graph neural networks (Sanchez-Gonzalez et al., 2020), Encoder-Process-Decoder architectures (Stachenfeld et al., 2022), and neural ODEs (Ayed et al., 2019).

Our proposed method sits between the last two categories. It inherits the advantages of both classes: our ansatz network is equally flexible compared to the free latent-spaces used in pure ML surrogates, and once trained there is no need so solve any optimization problem at inference time, in contrast to many neural-ansatz-type methods. In addition, our method is mesh-agnostic and therefore not subjected to mesh-induced stability constraints, which are typical in classical PDE solvers and their ML-enhanced versions. Given that we seek to build surrogate models for trajectories of a potentially *unknown* underlying PDE, we do not intend to replace nor compete with traditional PDE solvers.

## 3 METHODOLOGY

### 3.1 MAIN IDEA

We consider the space-time dynamics of a physical system described by PDEs of the form

$$\begin{cases} \partial_t u(x,t) & = \mathcal{F}[u(x,t)], \\ u(x,0) & = u_0. \end{cases} \tag{1}$$

---

**Algorithm 1** Approximating $u(x, T)$

---

**Input: initial condition:** $u_0$, **time horizon:** $T$
1. Construct $\mathbf{u}_0$ following y sample or interpolating $u_0$ to an equi-spaced grid $\{x_i\}_{i=0}^{N-1}$
2. Compute initial latent-space condition $\theta_0 = \Lambda_\alpha(\mathbf{u}_0)$
3. Compute $\theta(T)$ by integrating $\dot{\theta} = \Gamma_\beta(\theta)$.
4. Decode the approximation $\mathbf{u}(x, T) \approx \Psi_{\theta(T)}(x)$
5. Sample the approximation to the grid $\mathbf{u}^T = \mathbf{\Psi}(\theta(T))$
**Output:** $\mathbf{u}^T$

---

where $u : \Omega \times \mathbb{R}^+ \to \mathbb{R}^d$ denotes the state of the system with respect to spatial location $x$ and time $t$, $\mathcal{F}[\cdot]$ is a time-independent and potentially nonlinear differential operator dependent on $u$ and its spatial derivatives $\{\partial_x^\alpha u\}_\alpha$, and $u_0 \in \mathcal{U}_0$ is the initial condition where $\mathcal{U}_0$ is a prescribed distribution. For a time horizon $T$, we refer to $\{u(x, t),$ s.t. $t \in [0, T]\}$ as the *rollout* from $u_0$, and to the partial function $u(\cdot, t^*)$ with a fixed $t^*$ as a *snapshot* belonging to the function space $\mathcal{U} \supset \mathcal{U}_0$. We call $\mathcal{U}$ the *ambient space*.

**Encoder-Decoder Architecture**   For any given $t$ we approximate the corresponding snapshot of the solution $u$ as

$$u(x, t) = \Psi_{\theta(t)}(x), \tag{2}$$

where $\Psi$ is referred to as the *ansatz* decoder: a neural network tailored to the specific PDE to solve, so it satisfies boundary conditions and general smoothness properties, cf. Eq. (10). The *latent variables* $\theta(t) \in \Theta$ are the weights of $\Psi$ and are time-varying quantities. $\Theta$ is referred to as the *latent space*.

To compute the *latent variables* $\theta(t)$, we use a hypernetwork encoder, which takes samples of $u$ on a grid given by $x_i = i\Delta x$ for $i = 0, ..., N$, and outputs:

$$\theta(t) = \Lambda_\alpha \left( \{u(x_i, t)\}_{i=0}^N \right), \tag{3}$$

such that Eq. (2) is satisfied. Here $\alpha$ encompasses all the trainable parameters of the encoder.

**Latent Space Dynamics**   We assume that $\theta$ satisfies an ODE given by

$$\begin{cases} \dot{\theta}(t) & = \Gamma_\beta(\theta(t)), \\ \theta(0) & = \Lambda_\alpha(\{u_0(x_i)\}_{i=0}^N) \end{cases} \tag{4}$$

where $\Gamma$ is another hypernetwork whose trainable parameters are aggregated as $\beta$.

Algorithm 1 sketches how to solve PDEs through the latent space. For a given initial condition $u_0$, we uniformly sample (or interpolate unstructured samples) on a grid and compute its latent representation using the encoder via Eq. (3). We numerically solve the ODE in Eq. (4) to obtain $\theta(T)$. We then use the ansatz decoder to sample $u(x, T) \approx \Psi_{\theta(T)}(x)$, at any spatial point $x$. This is reflected in the blue computation path in Fig. 1.

To simplify the presentation, we consider only one trajectory, and assume that the sampling grid, denoted by $\mathbf{x} = \{x_i\}_{i=0}^N$, is fixed. Then $\mathbf{u}(t)$ is used to represent the vector containing the samples of $u(\cdot, t)$ on $\mathbf{x}$, or $[\mathbf{u}(t)]_i = u(x_i, t)$. Further, we conveniently denote $\mathbf{u}_t = \mathbf{u}(t)$ and $\theta_t = \theta(t)$. Thus, following this notation we have

$$\theta_t = \Lambda_\alpha(\mathbf{u}_t), \tag{5}$$

and $\mathbf{\Psi}(\theta)$ denotes the vector approximation of $u$ sampled on $\mathbf{x}$, i.e., $[\mathbf{\Psi}(\theta_t)]_i = \Psi_{\theta_t}(x_i)$.

**Consistency Regularization from Commutable Computation**   Our goal is to identify the structure of $\Psi$, $\Lambda$ and $\Gamma$, together with the parameters $\alpha$ and $\beta$ so the diagram in Fig. 1 commutes. In particular, we desire that the evolution in latent-space is equivalent to solving the PDE directly (denoted by the red path), but more computationally efficient.

Thus, a 'perfect' model of latent dynamics due to the encoder-decoder transformations should satisfy

$$\mathbf{u}_t = \mathbf{\Psi}(\Lambda(\mathbf{u}_t)), \qquad \dot{\mathbf{u}} = \nabla_\theta \mathbf{\Psi}(\theta)\dot{\theta}, \qquad \text{and} \qquad \dot{\theta} = \nabla_u \Lambda(\mathbf{u})\dot{\mathbf{u}}. \tag{6}$$

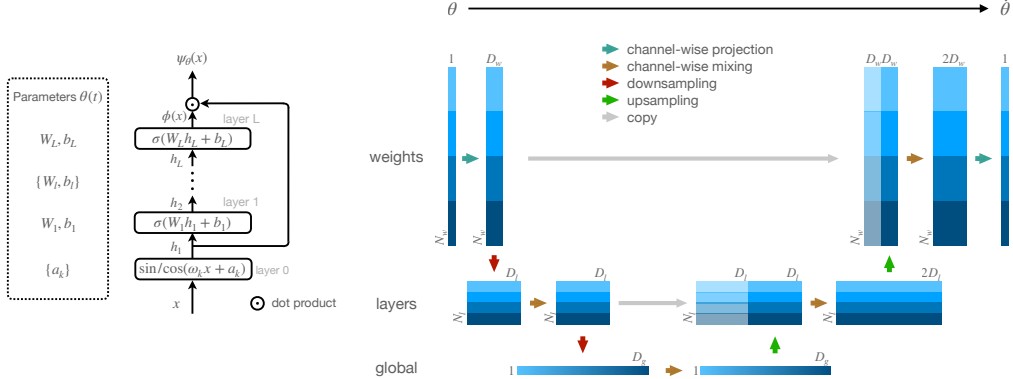

Figure 2: The Nonlinear Fourier Ansatz (left) featurizes the input with $\sin/\cos$ functions with specified frequency and trainable phases, and integrates a small MLP to parametrize functions which are periodic in $[0, L]$. The Hyper U-Net (right) architecture partitions weights according to their layers (shown in different shades) and extract/aggregate features similarly to the original U-Net. More details on the transforms represented by arrows in the Hyper U-Net are included in Appendix A.3.

However, imposing the conditions entails the complete knowledge of the time derivative in the ambient space, which may not be feasible. To build the encoder/decoder independently of the time derivatives, note that we can enforce the following constraints by combining the equations above:

$$\dot{\mathbf{u}} = \nabla_\theta \boldsymbol{\Psi}(\theta) \nabla_u \Lambda(\mathbf{u}) \dot{\mathbf{u}} \qquad \text{and} \qquad \dot{\theta} = \nabla_u \Lambda(\mathbf{u}) \nabla_\theta \boldsymbol{\Psi}(\theta) \dot{\theta}. \tag{7}$$

These expressions are the time derivatives of

$$\mathbf{u}_t = \Psi(\Lambda(\mathbf{u}_t)) \rightarrow \mathcal{L}_{\text{reconstruct}}(\mathbf{u}) = \|\mathbf{u}_t - \Psi(\Lambda(\mathbf{u}_t))\|^2 \tag{8}$$

$$\theta_t = \Lambda(\Psi(\theta_t)) \rightarrow \mathcal{L}_{\text{consistency}}(\theta) = \|\theta_t - \Lambda(\Psi(\theta_t))\|^2 \tag{9}$$

where the first equation leads to the typical reconstruction loss for autoencoders. The second equation, on the other end, leads to the *consistency regularization* which imposes the additional constraint $\theta = \Lambda(\Psi(\theta))$ whenever possible.

This symmetric pair of loss functions is depicted in Fig. 1 left. To enable transforming back and forth between ambient and latent spaces with little corruption, re-encoding $\mathbf{u}_t$ gives back $\theta_t$ is a must if we would take either red, blue or a mixed path to rollout the trajectory from any point in time.

## 3.2 ARCHITECTURAL CHOICES

We provide a succinct overview of the architectures used for the ansatz, encoder and dynamics.

**The Nonlinear Fourier Ansatz (NFA)** As an example, in this work we consider an ansatz applicable to a wide range of advection-dominated systems:

$$\Psi_\theta(x) = \sum_{k=1}^{K} \phi_k(x) \sin(\omega_k x + a_k) + \phi_{-k}(x) \cos(\omega_k x + a_{-k}), \tag{10}$$

where $\{\omega_k\}$ follow either a linear or a dyadic profile (see Appendix A.3). This ansatz leverages low-frequency Fourier modes as an envelop, corrected by a shallow neural network in their amplitude and phase (the architecture is depicted in Fig. 2). The ansatz is periodic in $[0, L]$ by construction, and can be further tailored to specific applications by adjusting the number of Fourier modes and their frequencies. In this case, $\theta$ aggregates the phases $a_k$ and the parameters inside MLPs in $\phi_k$.

**Encoder** Since we assume that the input data of the encoder lies in a one-dimensional equispaced grid, and that the boundary conditions are periodic, we use a simple convolutional encoder with periodic padding. This is achieved by a sequence of convolutional ResNet modules (He et al., 2016) with batch normalization (Ioffe & Szegedy, 2015) and a sinusoidal activation function, in which the input is progressively downsampled in space. We consider either four or five levels of downsampling. Each

level consists of two consecutive ResNet modules and then we downsample the spatial component by a factor of 2 while doubling the number of channels, thus maintaining the same amount of information throughout the layers. A fully connected layer at the output generates the parameters of the ansatz network.

**Hyper U-Net for Learning Dynamics** Since the input and output of function $\Gamma$ both follow the weight structure of the same neural network $\Psi$, we propose a novel neural network architecture, which we refer to as the Hyper U-Net, to model the mapping between them (illustrated in Fig. 2). Analogous to the classical U-Net (Ronneberger et al., 2015), which has 'levels' based on different resolutions of an image, the Hyper U-Net operates on 'levels' based on different resolutions of the computational graph of the ansatz network. In this paper, we use three levels which are natural to modern neural networks, namely individual weights, layer and graph levels, listed in decreasing order of resolution. At each level, features are derived from the finer level in the downsampling leg of the U-Net, and later on mixed with the ones derived from coarser level in the upsampling leg. Differently from the U-Net, instead of convolutions we use locally-connected layers that do not share weights.

## 3.3 Multi-stage Learning Protocol

Joint training of encoder, decoder and the dynamics U-Net is difficult, due to the covariate shift among these components (ablation study in Appendix A.5). To overcome this, we propose an effective multi-stage learning procedure.

**Learning to Encoder Snapshots** For a given architecture of an ansatz network, we learn an encoder $\Lambda_\alpha$ that maps snapshots to the corresponding latent spaces, by optimizing the loss

$$\mathcal{L}_{\text{enc}}(\alpha) = \sum_n \left[ \mathcal{L}_{\text{reconstruct}}(\mathbf{u}_{t_n}) + \gamma \mathcal{L}_{\text{consistency}}(\theta_{t_n}) \right]. \tag{11}$$

The relative strength between the two terms in the loss function, is controlled by the scalar hyperparameter $\gamma$. We found empirically that the second loss term injects inductive bias of preferring smoother and better-conditioned trajectories for the dynamic models to learn (see results section).

**Learning to Evolve the Latent Representation** After the encoder is trained, we generate training latent space trajectories by computing Eq. (3) on all $\mathbf{u}$. We then learn the latent dynamics operator $\Gamma_\beta$ from sample latent rollouts $\mathcal{D}_\theta = \{(t_0, \theta_{t_0}), (t_1, \theta_{t_1}), ...\}$ via the following loss function:

$$\mathcal{L}_{\text{ODE}}(\beta) = \sum_n \left\| \theta_{t_n} - \tilde{\theta}_{t_n} \right\|^2, \quad \tilde{\theta}_{t_n} = \theta_{t_0} + \int_{t_0}^{t_n} \Gamma_\beta(\theta(t)) \, dt, \tag{12}$$

where we use an explicit fourth-order Runge-Kutta scheme to compute the time integration. As our encoder identifies latent spaces with good smoothness conditions, we have the luxury to use relatively large, fixed step sizes, which serve as an additional regularization that biases towards learning models with efficient inference. Learning $\Gamma_\beta$ directly using this latent-space rollouts is empirically challenging, due to a highly non-convex energy landscape and gradient explosions. Instead, we split the training process in two stages: single-step pre-training, and fine-tuning using checkpointed neural ODE adjoints (Chen et al., 2018). We describe them briefly in below:

*Single-step pretraining* We train the latent dynamical model by first optimizing a single-step loss:

$$\mathcal{L}_{\text{single}}(\beta) = \sum_n \left\| \frac{\theta_{t_{n+1}} - \theta_{t_n}}{t_{n+1} - t_n} - \Gamma_\beta(\theta_{t_n}) \right\|^2, \tag{13}$$

which simply attempts to match $\Gamma_\beta$ with the first order finite difference approximation of the time derivative in $\theta$. Note that cumulative errors beyond the $(t_n, t_{n+1})$ window do not affect the learning of $\beta$.

*Multi-step fine-tuning* Although learning with single-step loss defined by Eq. (13) alone has been demonstrated to achieve success in similar contexts when optimized to high precision and combined with practical techniques such as noise injection (Pfaff et al., 2020; Han et al., 2022), we empirically observe that additional fine-tuning $\beta$ using Eq. (12) as the loss function helps a great ideal in further improving the accuracy and stability of our *inference rollouts*, i.e. it is important to teach the model

to "look a few more steps head" during training. The fact that our encoders generate smooth latent spaces, together with the use of fixed, large-step time integration, helps to reduce the cost of the otherwise expensive adjoint gradient compute, which had been one of the main drawbacks of such long-range neural-ODE training. We point out that there exists ample literature on multi-step training of differential equations, e.g. Keisler (2022); Brandstetter et al. (2022). For problems considered in this work, however, this simple two-stage training proved empirically effective.

To conclude this section, we mention in passing that the recently proposed Neural Galerkin approach (Bruna et al., 2022) can be used to learn the dynamics by fitting $\dot{\theta}$:

$$\partial_t u(x, \cdot) = \partial_t \Psi_{\theta(\cdot)}(x) = \nabla_\theta \Psi(x)\dot{\theta} = \mathcal{F}[u](x). \tag{14}$$

However, as demonstrated in our experiments, this approach can be at times prone to bad conditioning in the stiffness matrix, and relies on having complete knowledge about $\mathcal{F}$.

## 4 RESULTS

### 4.1 SETUP

**Datasets** We demonstrate the efficacy of the proposed approach on three exemplary PDE systems: (1) Viscid Burgers (VB) (2) Kuramoto-Sivashinsky (KS) and (3) Kortweg-De Vries (KdV). These systems represent a diverse mix of dynamical behaviors resulting from advection, characterized by the presence of dissipative shocks, chaos and dispersive traveling waves respectively. Each is simulated on a periodic spatial domain using an Explicit-Implicit solver in time coupled with a pseudo-spectral discretization in space. We used the spectral code in `jax-cfd` (Dresdner et al., 2022) to compute the datasets. This solver provides very accurate and dispersion free ground-truth references. Detailed descriptions of the systems and the generated datasets can be found in Appendix A.1.

**Benchmark Methods** We compare to several recently proposed state-of-the-art methods for solving PDEs continuously in space and time: Neural Galerkin (NG) (Bruna et al., 2022) (at high and low sampling rates), DeepONet (Lu et al., 2021), Fourier Neural Operator (FNO) (Li et al., 2020b) and Dynamic Mode Decomposition (DMD) (Schmid, 2010). We also investigate variants of our model that uses the same encoder architectures but a convolution-based decoder and comparable latent space dimensions (abbreviated NDV - neural decoder variant). Details of those methods are in Appendix A.4.

**Metrics** For each system, we use a training set of 1000 trajectories with at least 300 time steps each. For evaluation, trained models are then used to generate multi-step rollouts on 100 unseen initial conditions. The rollouts are compared to the ground truths, which are obtained by directly running a high-resolution pseudo-spectral solver on the evaluation initial conditions. To quantitatively assess the predictive accuracy of the rollouts, we compute the point-wise relative root mean squared error (relRMSE)

$$\text{relRMSE}(\mathbf{u}^{\text{pred}}, \mathbf{u}^{\text{true}}) = \frac{\|\mathbf{u}^{\text{pred}} - \mathbf{u}^{\text{true}}\|}{\|\mathbf{u}^{\text{true}}\|} \tag{15}$$

as a measure for the short-term prediction quality. We also compare the spatial energy spectra (norm of the Fourier transform) of the predictions to that of the ground truths via the log ratio:

$$\text{EnergyRatio}_r(f_k) = \log\left(E_r^{\text{pred}}(f_k)/E_r^{\text{true}}(f_k)\right),$$
$$E_r(f_k = k/L) = \left|\sum_{i=0}^{N-1} u(x_i, t_r) \exp\left(-j2\pi k x_i/L\right)\right|, \tag{16}$$

where $L = N\Delta x$ and $j^2 = -1$. $r$ denotes the range of the prediction. This metric (0 for perfect prediction and positive/negative values mean over-/under-predicting energy in the corresponding frequencies) provides a translation-invariant assessment of the frequency features of the long-term predictions. Lastly, we record the inference cost of each method in terms of the wall clock time (WCT) taken and the number of function evaluations (NFE) per simulation time unit.

Table 1: Whether the benchmark methods are able to produce a working inference model for each system. *Working* is defined as exceeding 100% relRMSE in 40 rollout steps (twice the training range for our model).

| | VB | KS | KdV |
|---|---|---|---|
| Ours-NFA | ✓ | ✓ | ✓ |
| Ours-NDV | | ✓ | ✓ |
| NG-Lo | ✓ | | |
| NG-Hi | ✓ | ✓ | |
| DeepONet | ✓ | | |
| FNO | ✓ | ✓ | |
| DMD | ✓ | | |

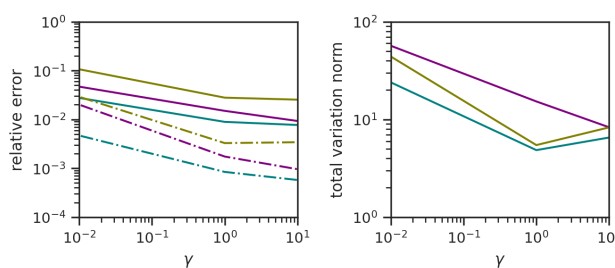

Figure 3: Relative reconstruction error (left, solid), consistency error (left, dashed) and total variation norm (right) vs. $\gamma$ at different learning rates for KS: $3 \times 10^{-4}$ —; $1 \times 10^{-4}$ —; $3 \times 10^{-5}$; —. See metric definitions in Appendix A.5.

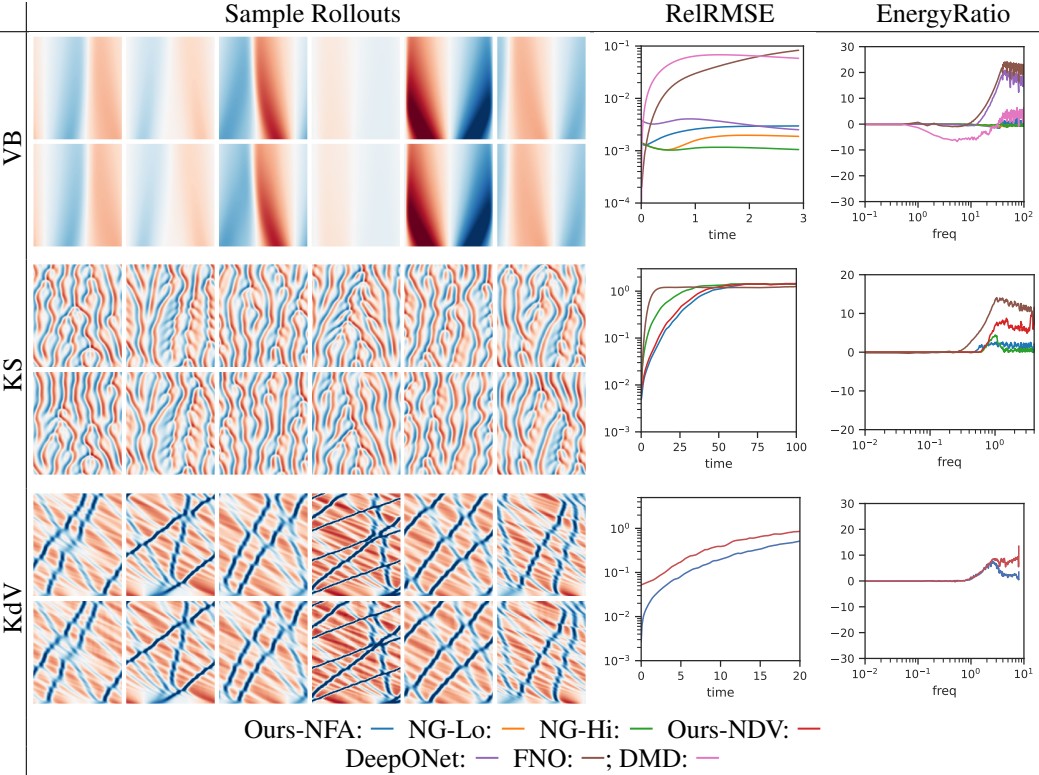

Figure 4: Sample long-term rollouts (left), point-wise error vs. time and long-range energy spectrum log ratios (right) plots. Rollouts are plotted in $(x, t)$ plane with upper rows showing the ground truths and lower rows our corresponding predictions. Columns correspond to different initial conditions. Ranges for energy ratios plotted: VB - 3; KS - 80; KdV - 16. NG-Lo/NG-Hi are NG runs with a sampling rate of 198 and 600 respectively.

## 4.2 MAIN RESULTS AND ABLATION STUDIES

**Accurate and stable long-term roll-outs** Fig. 4 contrasts our method to the benchmarks. While each benchmarked method fails to produce a working surrogate for at least one of the systems (defined and summarized in Table 1), our method is able to model all three systems successfully.

Qualitatively, shocks in VB, chaos patterns in KS, and dispersive traveling waves in KdV are all present in our model rollouts in ways that are visually consistent with the ground truths. Our models produce the lowest point-wise errors among working benchmarks for both KS and KdV (which are

Table 2: Comparison of inference speeds in terms of WCT and NFE statistics. WCTs are measured in miliseconds. NFEs are measured by an adaptive-step integrator. Numbers are normalized to cost per unit simulation time. Solver is run at the same resolution used to generate training data. High NFE count of NG-Hi results from running into a stiff numerical regime before breaking down (around 120 rollout steps).

| | VB | | | | KS | | | | KdV | | | |
| --- | --- | --- | --- | --- | --- | --- | --- | --- | --- | --- | --- | --- |
| | WCT | | NFE | | WCT | | NFE | | WCT | | NFE | |
| | mean | std | mean | std | mean | std | mean | std | mean | std | mean | std |
| Solver | 8.4 | 0.5 | - | - | 5.2 | 0.3 | - | - | 1041 | 13.1 | - | - |
| Ours-NFA | 3.8 | 0.9 | 16.0 | 4.1 | 1.7 | 2.2 | 5.3 | 0.2 | 52.1 | 18.4 | 164.7 | 40.3 |
| Ours-NDV | - | - | - | - | 1.1 | 0.6 | 7.1 | 2.7 | 18.0 | 5.4 | 71.4 | 19.5 |
| NG-Lo | 30.2 | 8.5 | 27.5 | 10.8 | 167.9 | 21.4 | 114.7 | 13.3 | - | - | - | - |
| NG-Hi | 49.6 | 7.7 | 27.5 | 10.8 | 319.4 | 33.5 | 111.7 | 11.7 | 35670 | 29035 | 11651 | 9502 |
| FNO | 18.1 | 14.5 | 100.0 | 0.0 | 0.4 | 0.4 | 5.0 | 0.0 | - | - | - | - |
| DMD | 2.4 | 0.4 | 100.0 | 0.0 | - | - | - | - | - | - | - | - |

the more complex examples) and come very close to the best-performing NG model for VB. In the long-range prediction regime, our models are able to remain stable for the longest period of time and exhibit statistics which perfectly match the truth in low frequencies and are the best approximating one in high frequency components.

**The importance of ansatzes** The right choice of the ansatz plays two roles. Firstly, learning the encoder becomes easier when the key structures are already embedded in the decoder - this is supported by the observation that our model easily learns to encode the shocks in VB while NDV fails to do so even allowed to operate a larger latent space. Secondly, the ansatz reduces the spurious high-frequency energy components in the state, which is helpful in improving the stability of rollouts, as these energies are likely to accumulate in a nonlinear fashion. This is the main reason why our model is able to have better errors than the closely related NDV model in KS and KdV.

**Consistency regularization leads to smoothness** The impact of $\gamma$ in Eq. (11) is evident from the observation that (Fig. 3 left) as the strength of regularization increases the encoders resulting from minimizing the loss in Eq. (11) have lower reconstruction error. It also induces smoother latent weights trajectories, as measured in the component-wise total variation norm (Fig. 3 right). The smoothness is beneficial for downstream training to learn efficient dynamical models that permit taking large integration time steps.

**Inference efficiency** Statistics on the inference cost of each method are summarized in Table 2. Our models have the best efficiencies amongst the compared methods. Their low NFE counts confirm that generous time steps are being taken during time integration. Note that the aforementioned trajectory smoothness is not a sufficient condition to guarantee such efficiency, supported by the observation that the PDE-induced dynamics found by NG typically require more NFEs per unit time (can even run into prohibitively stiff regimes, like in KdV) despite starting from the same encoded initial conditions. The explicit large-step integration that we use to learn the latent dynamical models proves to be an effective regularization to alleviate this issue - the NDV models are also able to achieve efficient inference benefiting from the same factor. However, without the help of consistency regularization, this comes at a compromise in accuracy and robustness compared to our main models.

## 5 CONCLUSION

In this work we have introduced a method to learn reduced-order surrogate models for advection-dominated systems. We demonstrate that through specially tailored ansatz representations and effective regularization tools, our method is able to identify weight spaces with smooth trajectories and approximate the dynamics accurately using hyper networks. Experiments performed on challenging examples show that our proposed method achieves top performance in both accuracy and speed.

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

# A APPENDIX

## A.1 DATASETS

In this paper, we consider PDEs of the form

$$\partial_t \boldsymbol{u} = \mathbf{F}(\boldsymbol{u}) := D\boldsymbol{u} + N(\boldsymbol{u}), \tag{17}$$

plus initial and boundary conditions. Here $D$ is a linear partial differential operator, and $N$ is a nonlinear term. For the models considered in this paper, the differential operator is typically either diffusive, $D = \partial_x^2$, dispersive, $D = \partial_x^3$, or hyper-diffusive, $D = \partial_x^4$, ; and the nonlinearity is a convective term, $N = \frac{1}{2}\partial_x(\boldsymbol{u}^2) = \boldsymbol{u}\partial_x\boldsymbol{u}$. Roughly, diffusion tends to *blur out* the solution, rendering the solution more homogeneous; dispersion produces *propagating waves* whose velocity depends on the Fourier content of the solution; finally, advection also propagates waves whose velocity depends on the local value of the solution.

In practice, PDEs are solved by discretizing space and time, which converts the continuous PDE into a set of update rules for vectors of coefficients to approximate the state $\boldsymbol{u}$ in some discrete basis, e.g., on a grid. For time-dependent PDEs, temporal resolution must be scaled proportionally to spatial resolution to retain an accurate and stable solution, so the runtime for PDE solvers scales at best like $O(n^{d+1})$, where $d$ is the number of spatial dimensions and $n$ is the number of discretization points along any dimension. In general, however, dispersion errors tend to dominate the pre-asymptotic regime so one would need a large amount of points in space, and in turn a large number of time steps.

To circumvent this issue, we choose a pseudo-spectral discretization, which is known to be dispersion free, due to the *exact* evaluation of the derivatives in Fourier space, and it possesses excellent approximation guarantees (Trefethen, 2000). Thus, relatively few discretization points are needed to represent solutions that are smooth.

We assume that, after appropriate rescaling in space, $\boldsymbol{u}(x,t) : [0, 2\pi] \times \mathbb{R}^+ \to \mathbb{R}$ is one-dimensional, $2\pi$-periodic and square-integrable for all times $t$. Consider the Fourier coefficients $\widehat{\boldsymbol{u}}^t$ of $\boldsymbol{u}(x,t)$, truncated to the lowest $K + 1$ frequencies (for even $K$):

$$\widehat{\boldsymbol{u}}^t = (\widehat{\boldsymbol{u}}_{-K/2}^t, \ldots, \widehat{\boldsymbol{u}}_k^t, \ldots, \widehat{\boldsymbol{u}}_{K/2}^t) \quad \text{where} \quad \widehat{\boldsymbol{u}}_k^t = \frac{1}{2\pi} \int_0^{2\pi} \boldsymbol{u}(x,t) e^{-ik \cdot x} dx. \tag{18}$$

$\widehat{\boldsymbol{u}}^t$ is a vector representing the solution at time $t$, containing the coefficients of the Fourier basis $e^{ikx}$ for $k \in \{-K/2, \ldots, K/2\}$. The integral $\widehat{\boldsymbol{u}}_k^t$ is approximated using a trapezoidal quadrature on $K + 1$ points, i.e., sampling $\boldsymbol{u}(x,t)$ on an equi-spaced grid, and then computing the Fourier coefficients efficiently in log-linear time using the Fast Fourier Transform (FFT) (Cooley & Tukey, 1965). Differentiation in the Fourier domain is a diagonal operator. In fact, it can be calculated by element-wise multiplication according to the identity $\partial_x \widehat{\boldsymbol{u}}_k = ik\widehat{\boldsymbol{u}}_k$. This makes applying and inverting linear differential operators trivial since they are simply element-wise operations (Trefethen, 2000).

The nonlinear term $N(\boldsymbol{u})$ can be computed via an expensive convolution, or by using Plancherel's theorem to pivot between real and Fourier space to evaluate these terms in quasilinear time.

This procedure leads to a system in Fourier domain of the form

$$\partial_t \widehat{\boldsymbol{u}}^t = \mathbf{D}\widehat{\boldsymbol{u}}^t + \mathbf{N}(\widehat{\boldsymbol{u}}^t), \tag{19}$$

where $\mathbf{D}$ encodes the $D$ operator in the Fourier domain and is often a diagonal matrix whose entries only depend on the wavenumber $k$. $\mathbf{N}$ encodes the nonlinear part. This system may be solved using a 4th order implicit-explicit Crack-Nicolson Runge-Kutta scheme (Canuto et al., 2007). In this case, we treat the linear part implicitly and the nonlinear one explicitly.

We point out that this transformation still carries some of the difficulties of the original problem. In particular, as $k$ increases the conditioning number of $\mathbf{A}$ will also increase, which may require refining the time-step in order to avoid instabilities and to satisfy the Shannon-Nyquist sampling criterion.

The solver used to generate data is implemented using `jax-cfd`, where we use $512$ grid points.

**Viscid Burgers Equation** We solve the equation given by

$$\partial_t u + u\partial_x u - \nu\partial_{xx} u = 0 \qquad \text{in } [-1, 1] \times \mathbb{R}^+, \tag{20}$$

with periodic boundary conditions. We follow the set up in (Wang et al., 2021), in which $\nu = 0.01$, and the initial condition is chosen following a Gaussian process.

We point out that the combination of the high-viscosity and the smoothness of the initial conditions results in trajectories that are fairly stable with waves travelling only small distances. No sub-grid shocks are formed.

**Kuramoto-Sivashinsky Equation**    We solve the equation given by

$$\partial_t u + u \partial_x u + \nu \partial_{xx} u - \nu \partial_{xxxx} u = 0 \qquad \text{in } [0, L] \times \mathbb{R}^+, \tag{21}$$

with periodic boundary conditions, and $L = 64$. Here the domain is rescaled in order to balance the diffusion and anti-diffusion components so the solutions are chaotic (Dresdner et al., 2022).

The initial conditions are given by

$$u_0(x) = \sum_{k=1}^{3} \sum_{j=1}^{n_c} a_j \sin(\omega_j * x + \phi_j), \tag{22}$$

where $\omega_j$ is chosen randomly from $\{\pi/L, 2\pi/L, 3\pi/L\}$, $a_j$ is sampled from a uniform distribution in $[-0.5, 0.5]$, and phase $\phi_j$ follows a uniform distribution in $[0, 2\pi]$. We use $n_c = 30$.

**KdV Equation**    We solve the equation

$$\partial_t u - 6u \partial_x u + \partial_{xxx} u = 0 \qquad \text{in } [-L/2, L/2] \times \mathbb{R}^+, \tag{23}$$

with periodic boundary conditions, and $L = 32$. Here the domain is again rescaled to obtain travelling waves. We point out that this equation is highly dispersive, so an spectral methods is required to solve this equation accurately.

The initial conditions for the KdV equation also follow Eq. (22), but with $L = 32$ and $n_c = 10$. This choice of initial conditions allows for non-soliton solutions, with a few large wave packages being propagated together with several small dispersive waves in the background.

Note that to our knowledge this is first time that a ML method is able to compute a non-soliton solution of the KdV equation for long times.

## A.2    Additional Training and Evaluation Details

For all training stages, we use the Adam optimizer (Kingma & Ba, 2015) with $\beta_1 = 0.9$, $\beta_2 = 0.999$ and $\epsilon = 10^{-8}$.

**Ansatz evaluation**    Prior to carrying out the multi-stage learning protocol described in the main text, we first perform a preliminary evaluation of the ansatz network via batch fitting $\{\theta^n\}$ to a random selection of snapshots $\{u_i\}$ drawn from the dataset, i.e. for each $i$ we minimize the representation loss

$$\mathcal{L}_{\text{rep}}\left(\{\theta^n\}\right) = \frac{1}{N} \sum_{n}^{N} \|\mathbf{u}^n - \mathbf{\Psi}(\theta^n)\|^2, \tag{24}$$

which provides us with an assessment of the expressive power of the particular functional form of $\Psi$ with respect to the problem at hand. Note that this process, otherwise known as *auto-decoder training* (Park et al., 2019), can be done independently and we can easily compare amongst multiple potential candidates and rule out ones which are not suitable. In particular, as a rule of thumb, we exclude ansatz expressions that cannot achieve below 1% relative reconstruction RMSE. As an example, in Table 3 we show the metric sweep of different decoder configurations for KdV. Similar procedures are carried out to determine the ansatzes for VB and KS.

Table 3: RelRMSE of the decoder to approximate snapshots of the KdV training trajectories resulting from different hyperparameters by minimizing Eq. (24). In this case we consider a dyadic scaling of the frequencies. We mark in bold font relRMSE below one percent.

| Act. fun. | relu | | | sin | | | swish | | |
|---|---|---|---|---|---|---|---|---|---|
| Feats/No. freqs | 3 | 6 | 9 | 3 | 6 | 9 | 3 | 6 | 9 |
| $(4, 4)$ | 0.1095 | 0.0255 | 0.0151 | 0.0803 | 0.0164 | 0.0110 | 0.1198 | 0.0164 | **0.0071** |
| $(4, 4, 4)$ | 0.1010 | 0.0249 | 0.0142 | 0.0511 | 0.0129 | 0.0085 | 0.0799 | 0.0120 | **0.0059** |
| $(8, 8)$ | 0.0273 | 0.0120 | 0.0117 | 0.0343 | **0.0095** | 0.0118 | 0.0637 | **0.0094** | 0.0054 |
| $(8, 8, 8)$ | 0.0195 | 0.0101 | **0.0094** | 0.0178 | **0.0082** | 0.0107 | 0.0333 | **0.0066** | 0.0050 |
| $(16, 16)$ | 0.0113 | 0.0090 | 0.0124 | 0.0202 | 0.0100 | 0.0181 | 0.0381 | **0.0076** | 0.0073 |
| $(16, 16, 16)$ | **0.0095** | **0.0079** | **0.0096** | 0.0115 | **0.0099** | 0.0178 | 0.0186 | **0.0057** | 0.0054 |

Table 4: Training specifications for encoder learning. Learning rate (LR) follows an exponentially decaying schedule, i.e. multiplying by a factor after a fixed period of steps.

| System | $\gamma$ | Batch Size | Training Steps | Base LR | LR decay | Steps Per Decay |
|---|---|---|---|---|---|---|
| VB | $10^0$ | 16 | 8M | $10^{-4}$ | 0.912 | 160K |
| KS | $10^1$ | 32 | 4M | $3 \times 10^{-4}$ | 0.892 | 80K |
| KdV | $10^1$ | 16 | 4M | $3 \times 10^{-4}$ | 0.945 | 40K |

**Encoder training**   This step is performed by minimizng loss funciton $Eq.$ (11). It acts as another round of validation to help us see if a particular ansatz have sufficient accuracy *when being encoded*. This is a stronger condition than the above-mentioned auto-decoding evaluation since the encoder adds extra regularization. We again set the bar to be 1% relative reconstruction error, which works well for us empirically.

Notably, ansatzes that have trainable frequencies ($\omega_k$ in Eq. (10)) fail this validation despite passing the auto-decoder test. This suggests that having these extra degrees of freedom in the ansatz is not as useful as hard-coding the periodic boundary condition directly, in terms of encoder learning.

Of all ansatzes that pass the 1% criterion, we choose the one with the fewest parameters, corresponding to the ones displayed in Table 6.

The encoder training specifications are summarized in Table 4.

**Dynamics training**   As mentioned in the main text, a two-stage pretrain/fine-tune protocol is adopted. The training parameters are listed in Table 5. We additionally employ gradient clipping (scaling norm to 0.25) to help stabilize training.

**Device info**   All training and inference runs are performed on single Nvidia V100 GPUs.

## A.3   MODEL DETAILS

**Ansatzes**   The hyperparameters of the ansatz used for each system are summarized in Table 6. The columns are explained as follows:

- *features*: MLP hidden layer sizes

- *activation*: nonlinearities applied after each hidden layer of the MLP; swish (Ramachandran et al., 2018) or sin function

- *frequency profile*: if linear, $\{\omega_k\}$ follow an arithmetic sequence $\{\pi/L, 2\pi/L, 3\pi/L, ...\}$; if dyadic, the sequence is geometric $\{\pi/L, 2\pi/L, 4\pi/L, ...\}$ in powers of 2

- *frequency dimensions*: each non-zero frequency has associated sin and cos functions with distinct phases; the input and output dimensions of the MLP are both $2 \times$ No. Freqs $+$ **1**[Has Zero Freq].

Table 5: Training specifications for dynamics learning. Pretrain settings are the same for all systems. *Seq Length* refers to the number of steps over which to optimize the dynamical model parameters ($= 2$ for pretraining; $> 2$ for fine-tuning)

| Stage | Batch Size | Training Steps | Seq Length | Base LR | LR decay | Steps Per Decay |
|---|---|---|---|---|---|---|
| Pretrain | 32 | 4M | 2 | $10^{-4}$ | 0.955 | 40K |
| Fine-tune, VB | 32 | 1M | 10 | $3 \times 10^{-4}$ | 0.955 | 10K |
| Fine-tune, KS | 32 | 1M | 20 | $5 \times 10^{-5}$ | 0.955 | 10K |
| Fine-tune, KdV | 32 | 1M | 20 | $1 \times 10^{-5}$ | 0.955 | 10K |

Table 6: Hyperparameters for ansatz networks.

| System | MLP | | | Frequency | | No. Params |
|---|---|---|---|---|---|---|
| | Features | Activation | No. Freqs | Profile | Has Zero Freq | |
| VB | (4, 4) | swish | 3 | dyadic | No | 84 |
| KS | (8, 8) | sin | 3 | linear | No | 188 |
| KdV | (8, 8) | swish | 6 | dyadic | Yes | 297 |

**Encoder**    The encoder architecture consists of multiple levels of double (one downsampling and one regular) residual blocks. Each residual block follows the sequential layer structure: Conv $\rightarrow$ BatchNorm $\rightarrow$ sin activation $\rightarrow$ Conv $\rightarrow$ BatchNorm, where downsampling is applied via striding in the first Conv layer when applicable. Overall, each level downsamples the spatial dimension and increases the number of channels, both by a factor of 2. A standard fully-connected layer is applied at the end of the network to obtain the desired output dimension.

Encoder hyperparameters differ very little across systems, with VB encoder using 5 downsampling levels, compared to 4 used in KS and KdV (see Table 7).

**Hyper U-Net**    The transforms represented by the arrows in Fig. 2 are as follows:

- *channel-wise projection*: locally connected layer (with kernel size 1 and no bias); projects weights from 1 to $D_w$ dimensions (first layer) and back (last layer)

- *channel-wise mixing*: sequence of transformation Dense $\rightarrow$ Swish activation $\rightarrow$ Dense $\rightarrow$ Residual $\rightarrow$ LayerNorm applied to channels within the same layer (or to the global embedding channels); parameters are unshared among layers

- *downsampling*: locally connected layer compressing all weights channels within the same layer into a single layer embedding of dimension $D_l$; similarly, at the global level, all layer embeddings are compressed into a single global embedding of dimension $D_g$

- *upsampling*: the inverse of the downsampling transform mapping single size-$D_l$ embeddings back to multiple size-$D_w$ ones (also the size-$D_g$ global embedding into $N_l \times D_l$ layer embeddings)

- *copy*: similar to the original U-Net, channels from the downsampling stage are concatenated during the upsampling stage

Specifically for our ansatzes, we consider each MLP layer to encompass the weights and bias $\{W, b\}$ associated with the layer compute $\sigma(Wx + b)$. The phases $\{a_{\pm k}\}$ are considered a separate layer collectively.

Our Hyper U-Net architectures for different systems vary in the hyperparameters $\{D_w, D_l, D_g\}$. Their values are listed in Table 7.

## A.4    BENCHMARK MODEL DETAILS

### A.4.1    NEURAL GALERKIN

The Neural Galerkin approach (Bruna et al., 2022) seeks to compute evolution equations using calculus of variations in a time-continuous fashion.

Table 7: Hyperparameters for encoder and hyper U-Net

| System | Encoder | | Dynamical Model (Hyper U-Net) | | | |
|---|---|---|---|---|---|---|
| | Levels | No. Params | $D_w$ | $D_l$ | $D_g$ | No. Params |
| VB | 5 | 186,268 | 4 | 128 | 512 | 1,846,612 |
| KS | 4 | 218,116 | 4 | 512 | 1024 | 17,963,932 |
| KdV | 4 | 218,116 | 4 | 512 | 1024 | 17,963,932 |

At a high level, the Neural Galerkin method writes an evolution equation for the weights following:

$$M(\theta)\dot{\theta} = F(\theta), \tag{25}$$

where

$$M(\theta) = \int_0^L \nabla_\theta \Psi_\theta(x) \otimes \nabla_\theta \Psi_\theta(x) dx, \text{ and } F(\theta) = \int_0^L \nabla_\theta \Psi_\theta(x) \mathcal{F}(\nabla_\theta \Psi_\theta(x)) dx. \tag{26}$$

$\Psi$ is the neural ansatz as described in Eq. (10) and $\mathcal{F}$ is given by the underlying PDE in Eq. (17).

The matrix $M(\theta)$ is computed by sampling $\Psi_\theta(x)$. We employ uniform sampling on the domain to approximate the integral, namely,

$$M(\theta) = \frac{1}{N_{\texttt{samples}}} \sum_j^{N_{\texttt{samples}}} \nabla_\theta \Psi_\theta(x_j) \otimes \nabla_\theta \Psi_\theta(x_j). \tag{27}$$

We also use a trapezoidal rule to compute the integrals for the KdV equation, which works better for this particular equation.

In order to provide a fair comparison, we have attenuated two main issues with this method:

1. The ansatz, which is chosen *a priori*, may not be expressive enough to represent the trajectories fully. For a fair comparison we use the neural ansatz in Eq. (10) and with the same hyperparameters in Table 6, instead of the ones used in the original paper (Bruna et al., 2022).

2. $M(\theta)$ is often ill-conditioned, so we add a small shift $M_{\texttt{stab}}(\theta) = M(\theta) + \epsilon I$, for $\epsilon = 1e{-}4$. Even with with regularization the resulting ODE is still stiff (in our experiments the condition number is around $10^8$). Thus, to avoid this numerical issue, we perform the computation in double precision.

Finally, the initial condition needs to be optimized at inference time, which takes a considerable amount of time. In this case we use as an initial condition the condition given by our fully trained encoder

$$\begin{cases} M_{\texttt{stab}}(\theta)\dot{\theta} &= F(\theta), \\ \theta_0 &= \Lambda_\alpha(\mathbf{u}_0). \end{cases} \tag{28}$$

The method is implemented in JAX (Bradbury et al., 2018). We use the adaptive-step Dormand-Prince integrator (Dormand & Prince, 1980) implemented in `scipy.integrate.solve_ivp`, and we allow the routine to choose the adaptive time step. This choice allows us to identify when the ODE system becomes locally stiff.

As mentioned in the main text, we are able to compute the solutions for the VB and KS equations. However, for the KdV equation the ODE integration goes unstable in all of the trajectories. Even after playing with smaller tolerances, different regularization constant, and more accurate initial conditions, we were not able to avoid the stability issues.

### A.4.2 NEURAL DECODER VARIANT

In our main model, the ansatz network plays the role of a *decoder*, which is responsible for transforming from the latent to the ambient space. The biggest difference from a typical decoder is that the ansatz essentially *pins down* the latent space and as a result the decoding map is *not parametric*. In the neural decoder variant (NDV), we consider the *typical decoder* setup and implement a

Table 8: Specifications for NDV training.

| Stage | Batch Size | Training Steps | Seq Length | Base LR | LR decay | Steps Per Decay |
|---|---|---|---|---|---|---|
| Pretrain | 8 | 4M | 2 | $3 \times 10^{-4}$ | 0.95 | 40K |
| Fine-tune | 8 | 1M | 20 | $5 \times 10^{-5}$ | 0.95 | 10K |

convolution-based decoder, whose trainable weights parametrizes the decoding map. This allows us to contrast and study the impact of having a fixed-form ansatz that is tailored to the problem.

For NDV, we use exactly the same encoder (architecture and latent dimensions) as our main model. The decoder architecture is essentially the reverse of the encoder. It consists of a de-projection layer (reversing the last encoder layer), followed by a sequence of upsampling blocks (alternating regular and strided convolution layers), each of which increases the resolution and decreases the number of features by a factor of 2.

The latent dynamical model is a MLP with $(2048, 2048)$ features in its hidden layers. This corresponds to around 4.5M parameters, which is smaller compared to our hyper U-Net model. We did experiment with larger models but did not observe any performance gain.

We follow the same multi-stage protocol to train NDV. The specifications are summarized in Table 8.

We explored the use of consistency regularization on the NDV as well and did not observe any clear advantage in terms of reconstruction accuracy or smoothness gain. A possible explanation is that the latent space under NDV has the freedom to scale down and let the decoder change accordingly to maintain the same level of reconstruction performance. This results in lower values in the consistency regularization without necessarily improving the conditioning of latent trajectories.

For the VB system, we explored a myriad of hyperparameter combinations including layer depths/widths, activation functions and learning rates, none of which resulted in a model that can be considered generalizing well to all snapshots.

### A.4.3 DEEPONET

The DeepONet is a general framework to approximate operators using neural networks. In a nutshell, for an operator $\mathcal{G}$ between two function spaces such that

$$v = \mathcal{G}(w). \tag{29}$$

DeepONet seeks to approximate this map point-wise, i.e.,

$$G(w)(y) \approx v(y), \tag{30}$$

where $G$ is the DeepONet and $y$ are arbitrary points in the domain of $v$. Following the example shown above, DeepONets are usually split into two networks: the trunk network, whose inputs are samples of $w$, and the branch network, whose inputs are the evaluation points $y$.

In this case we use them to approximate the trajectory given by an initial condition $u_0$ at any point $(x, t)$ for $t > 0$ and $x$ in the domain. Namely we consider

$$u(x, t) \approx G(\mathbf{u}_0)(x, t) = \mathcal{N}_{\texttt{trunk}}(\mathbf{u}_0) \cdot \mathcal{N}_{\texttt{branch}}(x, t), \tag{31}$$

where $\mathcal{N}_{\texttt{trunk}} : \mathbb{R}^{N_{\texttt{samples}}} \to \mathbb{R}^m$ and $\mathcal{N}_{\texttt{branch}} : \mathbb{R}^2 \to \mathbb{R}^m$, are neural networks.

For both the trunk and branch networks we use fully-connected MLPs. We fit the model on the VB, KS and KdV datasets, each of which has $N_{\texttt{samples}} = 512$. For the VB data, we use rectified linear unit (ReLU) activation functions. For the KS and the KdV datasets, we found that $\tanh$ and Gaussian Error Linear Units (GELU) activation functions (Hendrycks & Gimpel, 2016) worked better, respectively.

For the branch network, we modified the input slightly. Instead of feeding $(x, t)$ directly, we compute periodic features for the spatial coordinate $x$ using sines and cosines at different frequencies. The main rationale for this is to impose periodicity in space in the network. Thus, the input to the branch

Table 9: FNO hyperparameters.

| System | $P$ | $K$ | Layer Width (SpectralConv) | Num Layers | Num Modes | Layer Width (Final Projection) |
|---|---|---|---|---|---|---|
| VB | 8 | 20 | 64 | 4 | 8 | 128 |
| KS | 32 | 20 | 256 | 4 | 16 | 128 |
| KdV | 32 | 20 | 256 | 5 | 16 | 128 |

network can be rewritten as

$$\tilde{x} = \begin{bmatrix} \sin(2\pi/Lx) \\ \cos(2\pi/Lx) \\ \sin(4\pi/Lx) \\ \cos(4\pi/Lx) \\ \sin(6\pi/Lx) \\ \cos(6\pi/Lx) \\ \sin(8\pi/Lx) \\ \cos(8\pi/Lx) \end{bmatrix}, \tag{32}$$

and the branch network is then given by

$$\mathcal{N}_{\texttt{branch}}(x,t) = \texttt{MLP}(\tilde{x}, t), \tag{33}$$

where MLP is a fully connected network. For the VB system, both the trunk and the branch network has 4 hidden layers with 256 neurons each. The full network has $528,896$ parameters in this case. For KS and KdV data, we use 5 layers with 512 neurons each. This resulted in a total of $2,369,536$ parameters.

The model was trained using the Adam optimizer (Kingma & Ba, 2015). After a hyper parameter sweep, the learning rate that provided the best results was $10^{-4}$ with an exponential decay every 10000 steps by a factor of $0.977$. We used a time horizon of 300 steps to train the networks in each case. We used 5000 epochs, with a batch size of $512$, in which each element of the batch is a combination of $\mathbf{u}_0$ and $(x, t)$.

### A.4.4 Fourier Neural Operator

The Fourier Neural Operator (FNO) (Li et al., 2020b) is an operator learning framework centered around the operator map/layer $\mathcal{K} : v \to v'$ defined as:

$$v'(x) = \int k(x, y)v(y) \, dy + Wv(x), \tag{34}$$

where $k(x, \cdot)$ and $W$ are parametrized by trainable parameters. The integral term (a.k.a. spectral convolution, or SpectralConv operation) is conveniently computed in the Fourier space, where low-pass filtering may be additionally applied (controlled by the number of modes). To make up the overall operator map, the input function is first raised to a higher dimension, and then goes through multiple FNO layers Eq. (34) interleaved with nonlinear activation functions, before finally projected to the desired output dimension using a local two-layer MLP.

In our benchmark experiments, we follow the training approach outlined in the original paper, using FNO to map from a stack of $P$ past snapshots $\{\mathbf{u}^{n-P}, ..., \mathbf{u}^{n-1}\}$ to $\mathbf{u}^n$, with the grid locations $x$ as an additional constant feature in the input. We train the model in an autoregressive manner by iteratively rolling out $K$ prediction steps, each step appending the current predicted snapshot to the input for predicting the next step (and removing the oldest to keep the input shape constant). L2 loss is averaged over all $K$ prediction steps and minimized with respect to the model parameters.

For each system, we performed sweeps over all hyperparameters. The best-performing ones are listed in Table 9.

### A.4.5 Dynamic Mode Decomposition

The Dynamic Mode Decomposition (DMD) method (Schmid, 2010) is a data-driven, equation-free method of modeling the spatiotemporal dynamics of a system. The key idea is to obtain a low-rank

linear operator which, when applied to the current state of the system, approximates the state of the system at the next timestep.

Suppose we are given the snapshots $\mathbf{u}^0, \mathbf{u}^1, \cdots, \mathbf{u}^{N_t}$, where $\mathbf{u}^k$ denotes the state at timestep $k$ (i.e. at time $k\Delta t$) and $N_t$ is the total number of time steps. We gather the input and output matrices, $\mathbf{U}$ and $\mathbf{U}'$ by stacking the each snapshot columnwise in $\mathbf{U}$ and the corresponding next timestep in $\mathbf{U}'$. Concretely,

$$\mathbf{U} = \begin{bmatrix} \mathbf{u}^0 \mid \mathbf{u}^1 \mid \mathbf{u}^2 \mid \cdots \mid \mathbf{u}^{N_t-1} \end{bmatrix} \quad \mathbf{U}' = \begin{bmatrix} \mathbf{u}^1 \mid \mathbf{u}^2 \mid \mathbf{u}^3 \mid \cdots \mid \mathbf{u}^{N_t} \end{bmatrix} \tag{35}$$

DMD learns a linear operator $\tilde{\mathbf{A}}$ of a specified rank $K$ such that the following quantity is minimized.

$$\|\tilde{\mathbf{A}}\mathbf{U} - \mathbf{U}'\|_F \tag{36}$$

where $\| \cdot \|_F$ denotes the Frobenius norm. Without the rank constraint, the quantity $\|\mathbf{A}\mathbf{U} - \mathbf{U}'\|_F$ is minimized by $\mathbf{A} = \mathbf{U}'\mathbf{U}^\dagger$, where $\mathbf{U}^\dagger$ denotes the Moore–Penrose pseudoinverse of $\mathbf{U}$. However, for large dimensionality and a large number of snapshots, it is intractable to compute the matrix $\mathbf{A}$ explicitly.

The DMD algorithm efficiently computes the approximate low rank operator $\tilde{\mathbf{A}} \approx \mathbf{A}$ by exploiting the singular value decomposition (SVD) of $\mathbf{U}$. See Schmid (2010) or Kutz et al. (2016) for more details on the DMD algorithm.

Since our datasets consist of multiple trajectories, we stack together the $\mathbf{U}$ and $\mathbf{U}'$ matrices from each trajectory to form the matrices $\mathbf{X}$ and $\mathbf{X}'$, respectively. For $N_{\texttt{traj}}$ trajectories consisting of $N_t$ time steps each, we compute the matrices $\mathbf{X}, \mathbf{X}' \in \mathbb{R}^{d \times M}$ where $M = N_{\texttt{traj}}(N_t - 1)$ and $d$ is the dimensionality of the data. For evaluation, an initial condition $\mathbf{u}^0 \in \mathbb{R}^d$ can be rolled forward to predict the future time steps $\hat{\mathbf{u}}^k$, $k > 0$, by repeated application of the operator $\tilde{\mathbf{A}}$.

For the VB, KS and KdV datasets, the number of trajectories $N_{\texttt{traj}}$ is 1024, 800 and 2048 respectively; and the number of time steps per trajectory $N_t$ is 300, 1200 and 800 respectively. The dimensionality of the data is $d = 512$ in each case. Following (Kutz et al., 2016), we choose the desired rank $K$ of the approximate operator $\tilde{\mathbf{A}}$ by inspecting the rate of decay of the singular values of the snapshot matrix $\mathbf{X}$. For the VB and KS datasets, we chose the rank to be 64. For the KdV dataset, we chose the rank to be 80. We used only 1024 of the 2048 KdV trajectories to avoid the DMD algorithm running out of memory.

## A.5 ABLATION STUDIES

**Consistency regularization** In Fig. 3, the relative reconstruction error is given by Eq. (15), and the relative consistency error is defined in a similar fashion as

$$\mathrm{relRMSE}_{\mathrm{consistency}}(\theta) = \frac{\|\theta - \Lambda \circ \mathbf{\Psi}(\theta)\|}{\|\theta\|}, \tag{37}$$

through which we measure whether we can recover the same $\theta$ after decoding and encoding again. The maximum component-wise, 1st order total variation norm is computed as

$$\max_i \frac{1}{Q} \sum_{q}^{Q} \sum_{n=0}^{N-1} \Delta t \left| \theta^{n+1,q} - \theta^{n,q} \right|, \tag{38}$$

where $q$ denotes index of evaluation cases and the maximum is taken over latent dimensions $i$. This quantity measures how fast/slow the latent variables changes over a fixed time window.

In Fig. 5, we show samples of latent trajectories obtained using various $\gamma$ values. We observe that trajectories under high $\gamma$ values are visibly *smoother*.

**Fine-tuning by multi-step training** In Fig. 6, we compare rollouts generated by the models obtained after the pretraining and fine-tuning stages respectively, to highlight the importance of the latter. We observe that for all systems fine-tuning is able to significantly reduce the point-wise errors and, in the case of KS and KdV, remove patches in the predictions that look rather 'unphysical'.

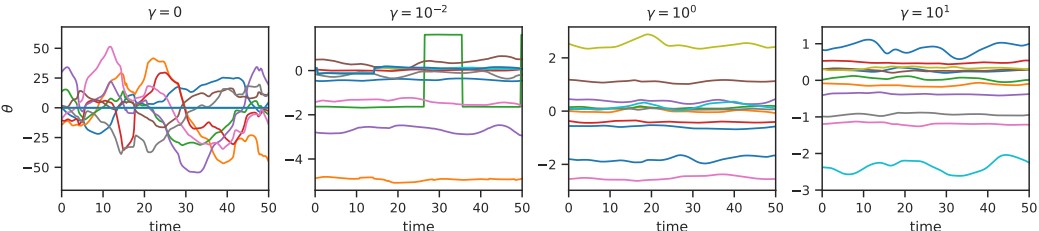

Figure 5: Latent trajectory samples resulted from training with different $\gamma$ values for KS system. Trajectories plotted are for the same 10 latent/weight dimensions (identified by the same colors) randomly selected for a fixed ansatz.

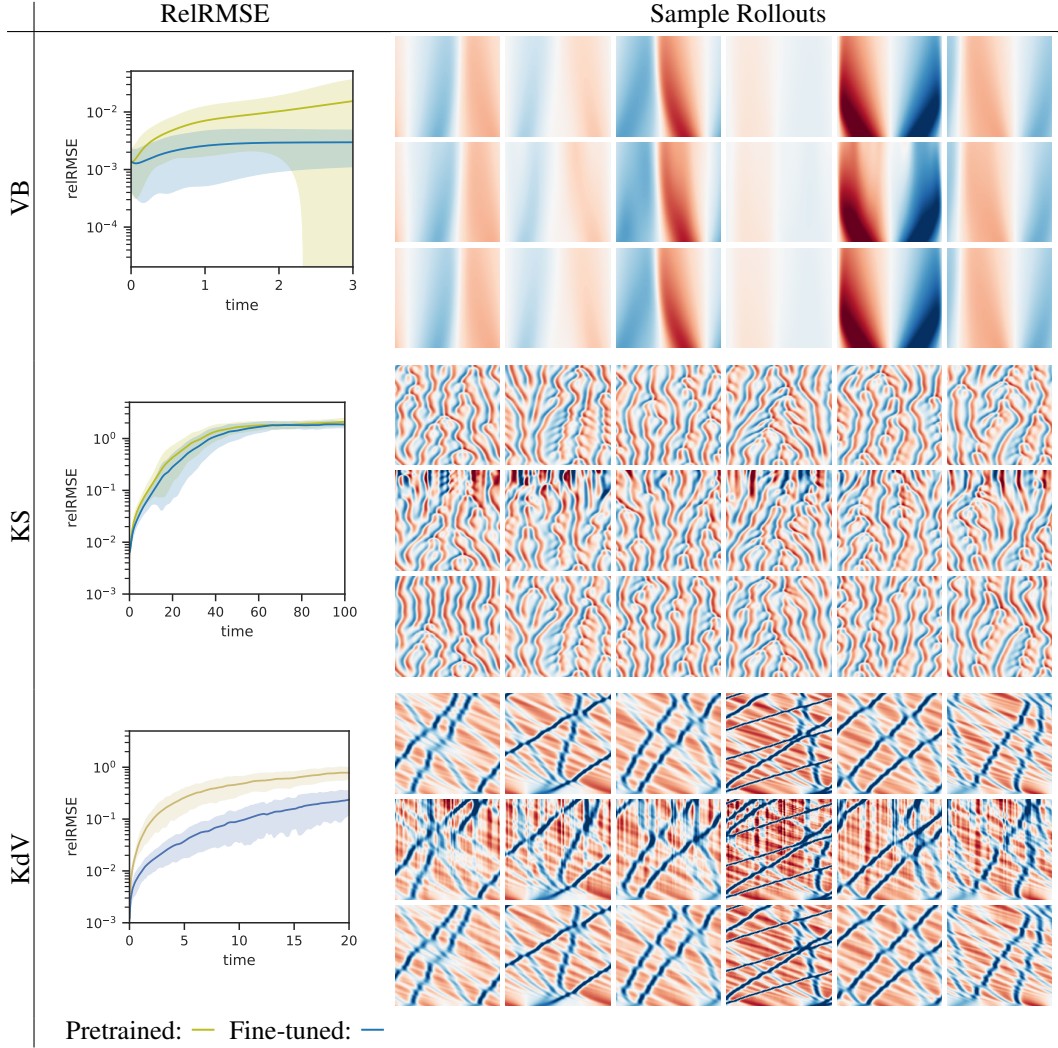

Figure 6: Relative RMSE (left) and sample rollouts (right) demonstrating the effects of fine-tuning with multi-step training. Rollouts are plotted in $(x, t)$ plane with three rows showing the ground truths, pretrained and fine-tuned predictions respectively.

**End-to-end (e2e) training**  In our multi-stage training protocol, we freeze our encoders when training the dynamical model. As an ablation study, we also attempted adding an additional multi-step, e2e training stage where the parameters for both the encoder and dynamical are allowed to change, optimizing for a single rollout loss in the ambient space. Despite observing steady decrease in

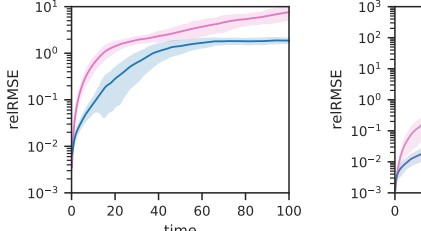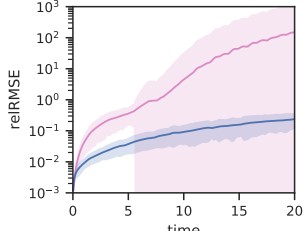

Figure 7: Relative RMSE for latent space fine-tuned (—) and e2e-trained (—) for KS (left) and KdV (right) systems.

Table 10: Mean relRMSE of the decoder to approximate snapshots of the KdV training trajectories Eq. (24). We consider a MLP with different number of features and activation functions. The number of layers for the first and last layer is chosen so that the total number of parameters is similar to using three frequencies.

| Features/Act. fun. | relu | sin | $\sin(\pi x)$ | swish | tanh |
|---|---|---|---|---|---|
| $(6, 4, 4, 6)$ | 0.6751 | 0.5585 | 0.0583 | 0.6395 | 0.6650 |
| $(6, 4, 4, 4, 6)$ | 0.6496 | 0.4208 | 0.0432 | 0.5343 | 0.5945 |
| $(6, 4, 4, 4, 4, 6)$ | 0.6420 | 0.3234 | 0.0837 | 0.4534 | 0.5362 |
| $(6, 8, 8, 6)$ | 0.4452 | 0.4291 | 0.0137 | 0.5653 | 0.5491 |
| $(6, 8, 8, 8, 6)$ | 0.3336 | 0.2325 | 0.0098 | 0.3953 | 0.4311 |
| $(6, 16, 16, 6)$ | 0.3260 | 0.3330 | 0.0092 | 0.4871 | 0.4211 |
| $(6, 16, 16, 16, 6)$ | 0.1873 | 0.1077 | 0.0092 | 0.2306 | 0.2268 |
| $(6, 32, 32, 6)$ | 0.2297 | 0.2634 | 0.0083 | 0.3768 | 0.2634 |
| $(6, 32, 32, 32, 6)$ | 0.0878 | 0.0494 | 0.0105 | 0.0873 | 0.0643 |
| $(6, 64, 64, 6)$ | 0.1229 | 0.2009 | 0.0091 | 0.2568 | 0.1086 |
| $(6, 64, 64, 64, 6)$ | 0.0424 | 0.0274 | 0.0117 | 0.0361 | 0.0214 |

the training loss, this process (perhaps counter-intuitively) was in reality found to hurt the quality of rollouts significantly (errors shown in Fig. 7), especially in its stability. We suspect the reason is that without enforcing the latent evolution to remain close to highly regularized the encoder output, it is easy for the optimizer to move to a nearby local minima with equal or potentially better performance in the ambient loss. It is highly likely that the resulting dynamics, on the other hand, is not amenable to long-range rollouts in the absence of proper regularization.

**Decoder ansatz** We benchmarked the approximation power of classical MLPs of different depth, width, and different activation functions. The benchmark is similar to the one in Table 3, in which we minimized Eq. (24) for trajectories of the KdV equation. For each experiment we solved the problem in Eq. (24) using SGD with a learning rate of $10^{-4}$, a batch size of $4$ with $0.5M$ training steps. For each configuration, this experiment was repeated for 1000 snapshots sampled from different trajectories of the KdV equation, and we computed the mean relative RMSE at the end of the optimization. The mean relative RMSE for each configuration is depicted in Table 10, in which we can observe that the errors are much higher than the ones reported in Table 3. In addition, Table 10 shows that imposing the periodicity condition into the activation function greatly helps to reduce the approximation error.

## A.6 ADDITIONAL METRICS

We provide some additional metrics including more spatial energy spectra at various prediction ranges (Fig. 8) and sample rollouts demonstrating a richer range of patterns which are well captured by our models (Fig. 9).

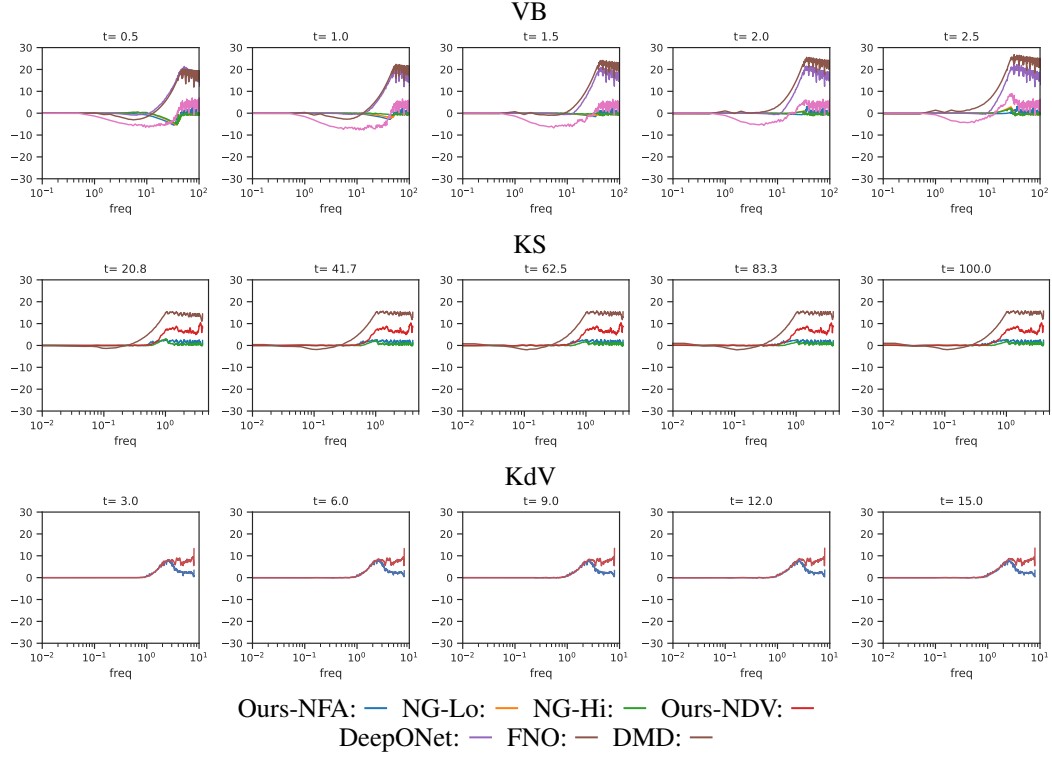

Figure 8: Spatial energy log ratio, computed using Eq. (16), for different prediction ranges.

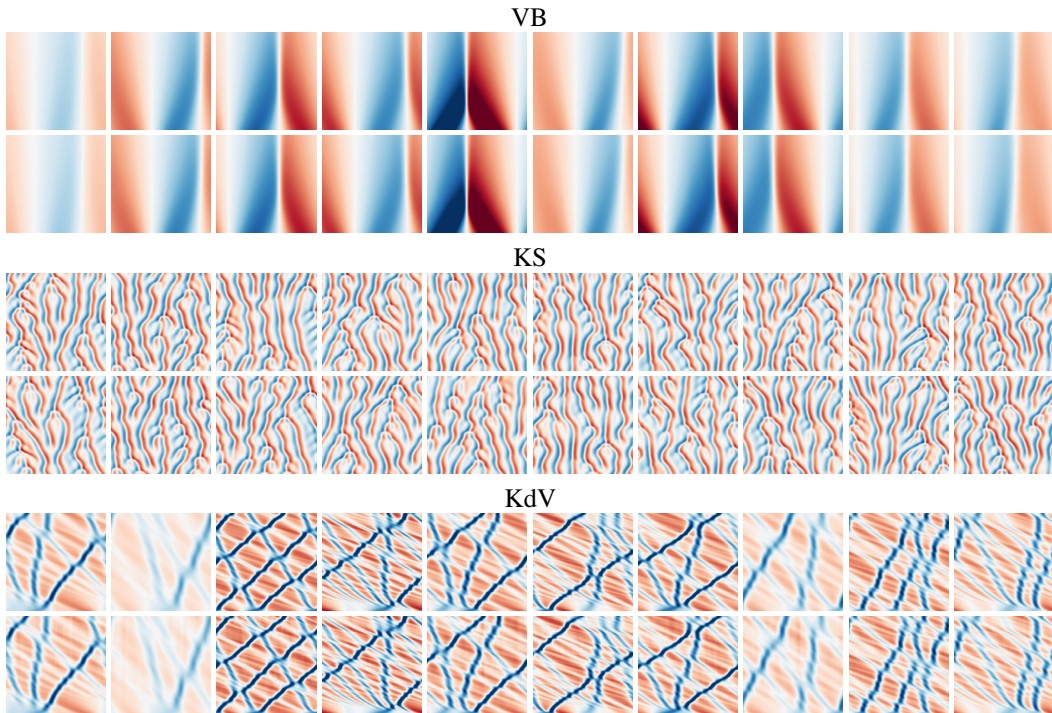

Figure 9: Additional evaluation rollouts plotted in $(x, t)$ plane. Top rows are ground truths and bottom rows are the corresponding predictions rolled out from the same initial conditions. Prediction range: VB - 3, KS - 80, KdV - 30.

