# OpenReview forum: "Evolve Smoothly, Fit Consistently: Learning Smooth Latent Dynamics For Advection-Dominated Systems"
_ICLR.cc/2023/Conference — ICLR 2023 notable top 25%_

### Official Review · Reviewer_mB8q · 2022-10-20

**Confidence:** 3
**Correctness:** 4
**Technical Novelty And Significance:** 3
**Empirical Novelty And Significance:** 3
**Recommendation:** 8

**Clarity, Quality, Novelty And Reproducibility:**


Clarity excellent. Quality excellent. Novelty high. Reproducibility ok.


**Strength And Weaknesses:**


s: this paper has excellent and effortless writing that demonstrates both clarity and expertise. I enjoyed reading this manuscript a lot! I also like how the paper makes does a transparent discussion of the training difficulties/annoyances

s: the idea is sensible, and turning the latent state into parameters of the decoder is elegant, which conceptually marries latent dynamics with operator learning. The construction of eqs 6-9 is very clever.

s: the results are good, and sufficiently elucidated


**Summary Of The Paper:**

The paper proposes latent PDE learning with novel autoencoder-type losses.


**Summary Of The Review:**


This is an excellent paper with refreshing, new ideas on this domain, good enough results and excellent writing.

---

> ### Author Response · Authors · 2022-11-13
> **Response to Reviewer mB8q**
>
> Thank you for the positive feedback! We are thrilled that you like our idea and enjoyed reading our paper!

---

### Official Review · Reviewer_haAi · 2022-10-21

**Confidence:** 2
**Correctness:** 4
**Technical Novelty And Significance:** 3
**Empirical Novelty And Significance:** 3
**Recommendation:** 8

**Clarity, Quality, Novelty And Reproducibility:**

Overall, the paper is very clear. I am not an expert in the field but it looks original enough even if the body of literature in latent space modeling is huge. It looks a bit difficult to reproduce the whole set of experiments, unless some care is taken regarding an accompanying github repo, which I could not check.

No typo I could spot. In the caption of table 2, replace "number of NFE" by just "NFE"

**Strength And Weaknesses:**

As a set of strength, I can say the paper is first useful for the not so knowledgeable reader to get references in the field. Then, the proposed method looks original to me, with its two nice contributions (cyclic losses, hypernet decoder). The method seems to be applicable even when the number of trajectories is quite limited (1000 here apparently). It was a nice read and I recommend it for publication.

As weaknesses, I could mention:
* The experiments look quite weak to me. I understand that great care was taken on this, but I basically wonder about real world usecases, which I cannot see here.
* I wonder about how critical is the choice of the decoder. The fact is that the chosen Fourier-based decoder looks pretty appropriate for the datasets at hand (figure 4), but it may be that the system would collapse for other kinds of dynamic data. Would there be some guidelines regarding what kind of decoder are appropriate or some ablation studies on that ?
* Likewise, even assuming a Fourier-based decoder, the choice of the frequencies look mysterious and I wonder whether their choice needs a lot of care. Why are they not trained ?
* Table 1 is not satisfying at all. I would have preferred some objective assessment of the encoder/decoder experiments, instead of this "working/not working" thing. What does "working" mean exactly anyways ?
* I don't understand the "spatial energy spectra" as a metric. Could you please explain it differently ?

**Summary Of The Paper:**

The paper proposes a method to model dynamical systems in a latent space in a smooth way. To go back and forth from the latent space, a decoder (called "ansatz") and an encoder network are used, that are task dependent. One first trick proposed in the paper is to propose some cyclic consistency for this encoder decoder, so that both latent->state->latent and state->latent->state should be identity functions. This is advocated to help for the smoothness of the latent variables.
Then, the core contribution of the paper in my view lies in the fact that the latent variables are not seen as an input to the decoder, but rather as encoding the *weights* of a network (in an hypernet way),  that inputs the coordinates (time, space) and outputs the value (nerf-style). This is advocated as yielding a much more powerful representation.
The rest of the paper is busy with describing important learning tricks and describing experiments.

**Summary Of The Review:**

I liked the paper and the approach looks promising to me, puting together some cyclic consistency idea for latent space modeling as well as nonparametric hypernet decoders.

---

> ### Author Response · Authors · 2022-11-13
> **Response to Reviewer haAi 1/2**
>
> We thank you for your time and suggestions.
>
> > The experiments look quite weak to me. I understand that great care was taken on this, but I basically wonder about real world use cases, which I cannot see here.
>
> The current experiments are chosen to capture the core difficulties of advection-dominated problems for which existing methodologies are insufficient. For example, the KdV equation has been notoriously difficult for surrogate modeling using ML-based techniques due to their nonlinear and dispersive nature, which makes the local speed of the solution depend not only on the local gradient but also on the local Fourier spectrum of the solution. To our knowledge no other ML methodology had been able to approximate the trajectories of such a seemingly innocuous equation.
>
> In this work we aim to provide evidence that using hypernetworks with a carefully constructed ansatz and regularization is a viable option for surrogate models for advection-dominated problems. Significant efforts are needed to extend our work to real world cases. We are actively working on applying our framework to the 2D Kolmogorov flow. This would be a prime example of a chaotic system with turbulent dynamics. We hope that such an example would bring the results closer to a more realistic use case. In addition, we will tackle more complex geometries and physics, hopefully using real data. However, this is out of the scope of the current manuscript.
>
> > I wonder about how critical is the choice of the decoder. The fact is that the chosen Fourier-based decoder looks pretty appropriate for the datasets at hand (figure 4), but it may be that the system would collapse for other kinds of dynamic data. Would there be some guidelines regarding what kind of decoder are appropriate or some ablation studies on that ?
>
> As you correctly point out, the choice of the decoder is critical. We did try several off-the-shelf architectures such as MLP with different activation functions and sizes, which all behaved rather poorly. In short, the Fourier-based discretization, which respects our periodical boundary conditions, performs significantly better.
>
> Alternatively, if the desired/target solution is not periodic or smooth, then it would be necessary to tailor an ansatz network for that solution. For example, if the solution needs to satisfy boundary conditions or is not periodic, an ansatz based on Chebyshev polynomials modulated by a neural network could be a better choice. However, note that our contribution is not about how to choose the best decoder architecture but also a generic learning framework for learning both the encoder and the decoder using the “hypernet” and the consistency loss.
>
> > Likewise, even assuming a Fourier-based decoder, the choice of the frequencies look mysterious and I wonder whether their choice needs a lot of care. Why are they not trained ?
>
> First, we want to note that having trainable frequencies does not strictly observe periodic boundary conditions (only integer frequencies do). Empirically, we found that hard-coding the BC outweighs having potentially more expressive powers by making frequencies trainable. This is reflected most in encoder learning. We added some explanations about this point in Appendix A.2.
>
> Secondly, we also know that we do not need very high frequencies (judging from how the snapshots look). Combined with the hard constraint of satisfying periodic BC, this leaves us only a handful of discrete choices, for which we can simply run a few evaluations, by fitting them to a random selection of snapshots (i.e. auto-decoder training), to determine the best choice. This is mentioned in Appendix A.2 under ‘ansatz evaluation’. In general the choice of frequencies is driven by having a small network with enough frequency content that it would be easy to fit the target solution. We have added Table 3 in the Appendix to show an example hyperparameter study for KdV. Following the vast literature in wavelets we considered a dyadic scaling of the frequency, which, in our experiments, provided the best tradeoff between accuracy and size of the decoder network.

---

> > ### Author Response · Authors · 2022-11-13
> > **Response to Reviewer haAi 2/2**
> >
> > > Table 1 is not satisfying at all. I would have preferred some objective assessment of the encoder/decoder experiments, instead of this "working/not working" thing. What does "working" mean exactly anyways ?
> >
> > We acknowledge your point. The main motivation for this table was to provide a qualitative assessment of the compared methods and prevent adding quantitative comparisons into Fig. 4 that would have skewed the scale and reduced the resolution of the graphs. In particular, the errors for methods that are labeled “not working” are accumulating much faster compared to others. However, we acknowledge that the criterion here should have been explicit. We have added such a concrete criterion (namely, the relative error exceeding 100% in 40 steps, which is twice the finetuning prediction length of our model) for leaving the results out of the graphs in Fig. 4 and communicated this in our revision.
> >
> > > I don't understand the "spatial energy spectra" as a metric. Could you please explain it differently ?
> >
> > The spatial energy spectrum is the square of the magnitude of the Fourier transform as a function of the frequency. In contrast with the pointwise error, this is a “translation-invariant” metric - meaning that two functions share the same spectrum as long as frequency projections have the same strengths (in other words, the precise locations, represented by the phases of the Fourier transform, do not matter). This metric is especially suitable for the advection-dominated problems that we consider here, where things are “pushed around the domain”. It complements the pointwise error metric in the sense that if we predict a bump on the left side of the domain while the truth contains a bump of the same shape on the right, we would still be considered to have made a good prediction under this metric where the pointwise metric would likely suggest the opposite.
> >
> > We have added a new sentence to provide more intuition behind this metric in our revision.

---

> > > ### Comment · Reviewer_haAi · 2022-11-28
> > > **thank you for your answers**
> > >
> > > the answers are satisfying to me and I maintain my good score.
> > >
> > > It still isn't clear in my view from equation (16) that the metric actually means comparing the resulting spectral energy. Reading the text, I understood you took E as the metric (which is just the magnitude Fourier transform of u), but it is actually |E - Etrue|, and this is where I was confused. Please add this in the revision.

---

> > > > ### Author Response · Authors · 2022-12-02
> > > > **Thanks for the feedback and suggestion**
> > > >
> > > > Thanks for the positive feedback regarding our replies.
> > > >
> > > > In Fig. 4 we actually do not show the difference in Fourier transform, but rather the Fourier transforms themselves (curves are meant to be compared against the ground truth labeled in color gray). On that note, however, we do agree that showing the difference may provide better visualizations. Thank you for the suggestion and we will update the Figure and text in the final version.

---

### Official Review · Reviewer_WMEw · 2022-10-23

**Confidence:** 3
**Correctness:** 4
**Technical Novelty And Significance:** 3
**Empirical Novelty And Significance:** 3
**Recommendation:** 6

**Clarity, Quality, Novelty And Reproducibility:**

__Clarity and Quality__:

The paper is generally well written and easy to follow. There are some minor issues as listed below:

In the introduction the authors say

> Those systems have slow-decaying Kolmogorov n-width that hinders standard methods,

However, I could not find any further details in the text?

- The word "Ansatz" is sometimes capitalized, sometimes not.

__Novelty__:

- I am not familiar enough with the literature to comment on novely.




**Strength And Weaknesses:**

__Strengths__:
- Interesting idea on how to make use of latent variables for dynamic data.
- The idea is well thought out in several aspects, e.g. different loss functions, simpler pretraining, the smooth dynamics etc.
- The performance seems good across tasks
- Comprehensive list of relevant work

__Weaknesses__:

- On a high level, the idea of producing latent trajectories to encode temporal data has been present for some time and was also discussed in the original neural ODE paper, which doesn't seem to be mentioned here. But it is also only a very high level connection.

**Summary Of The Paper:**

The paper introduces a latent representation of dynamic systems by proposing an autoencoder framework specialised to PDEs. The authors motivate and introduce several components for the loss functions such as reconstruction, consistency as well a single step pre-training. Good performance is demonstrated on several examples.

**Summary Of The Review:**

The paper has several positive aspects and seems to be thought out well. Empirical results are promising, so I am tending towards acceptance.

---

> ### Author Response · Authors · 2022-11-13
> **Response to Reviewer WMEw**
>
> We highly appreciate for your time and feedback.
>
> > On a high level, the idea of producing latent trajectories to encode temporal data has been present for some time and was also discussed in the original neural ODE paper, which doesn't seem to be mentioned here. But it is also only a very high level connection.
>
> Indeed, our method is connected to  neural ODEs on a high level. There are several notable distinctions: (a) we impose particular structures in the latent space (i.e. the weights of a chosen ansatz network) tailored to our specific problems (b) we employ a more effective training routine that is designed to improve the training efficiency as directly computing the adjoint on a long time series is very expensive and slow to converge. The latter point was noted in the method section of our first draft, with references to other previous work that uses similar techniques. We expanded it with more texts and also stressed the connection in the introduction section of our revision.
>
> > Those systems have slow-decaying Kolmogorov n-width that hinders standard methods,
> However, I could not find any further details in the text?
>
> We apologize for this oversight. We have added a new paragraph with a succinct description of the Kolmogorov n-width with the corresponding references in the text.
>
> > The word "Ansatz" is sometimes capitalized, sometimes not.
>
> Thank you for pointing it out. This has been addressed in our revision.

---

### Author Response · Authors · 2022-11-13
**General Response**

We thank all the reviewers for their time and comments. We have uploaded a revised manuscript, where the new texts are in blue color.  We are pleased that the ideas of hyper-net and consistency loss are well appreciated as they improve the model’s expressiveness, ease the learning process. We appreciate the comments about the reproducibility: we will release our codes, give pointers to baselines used in this paper, publish the datasets (and the codes for generating the datasets).

---

> ### Author Response · Authors · 2022-11-21
> **List of updates to the manuscript**
>
> We have updated the paper following the comments and suggestions of the reviewers. For the sake of clarity and easier comparison, the modifications are colored blue:
>
> * Page 1: we have added some background information on the Kolmogorov n-widths, the intuition behind them, and why they matter for projection-based ROMs.
> * Page 2, second paragraph: we stressed the link to neural ODEs and mentioned that we do leverage them in order to learn the latent dynamics.
> * Page 6: we further elaborated on the difference between our methodology and neural-ODE, particularly during the training stage.
> * Page 7: we modified Table 1 to provide a quantitative discriminator for whether different methods are considered to be "working". We have also added two more methods in our benchmarks, the relatively new Fourier neural operators (FNO), and the classical dynamical mode decomposition (DMD).
> * Page 7, last two paragraphs: we added a short discussion on the energy spectrum metric.
> * Page 8, Fig. 4: we added the two new methods to the graphs. We also cleaned the graphs to show only the mean, which would hopefully make the plots easier to read.
> * Page 16: we added Table 3 with some results on an example hyperparameter search of the encoder ansatz, and we added a section to describe how such a parameter search is performed.
> * Page 20: we added a section that gives some details on FNO, one of the new baselines, where we also provided details on the hyperparameters used for the benchmarks.
> * Page 21: we added a section that provides details about DMD, and the parameters we used for its implementation, particularly for the benchmarks.
> * Page 23: we added another ablation study for the particular choice of decoder architecture. We briefly explain the benchmark and we provide the results of such comparison in Table 10, which stands in contrast to Table 3 (introduced before).
>
> We also cleaned up the writing in general by improving some wording and removing typos, inconsistent spellings, etc.

---

> ### Public Comment · ~Shuhao_Cao1 · 2023-03-28
> **Nice paper**
>
> Nice paper. I have a minor question about the writing.
>
> - There are several occasions that the term "complete knowledge of the time derivative" or "complete knowledge of $\mathcal{F}$". What does this "complete knowledge" mean? Able to evaluate these operators at a given discretization?
>
> BTW: I wonder if the codes and datasets have been released yet.

---

> > ### Author Response · Authors · 2023-03-30
> > **Thanks for the inquiry.**
> >
> > "Complete knowledge" simply means that we know the exact function/operator form (without error) of $\mathcal{F}$ as in $\dot{u} = \mathcal{F}(u)$, in order words the full ODE/PDE expressions.
> >
> > We are still improving the quality of the code and will release it as soon as we can.

---

### Decision · Program_Chairs · 2023-01-20

**Decision:**

Accept: notable-top-25%

**Justification For Why Not Higher Score:**

The results are not so fundamental that they might need the attention of the entire community.

**Justification For Why Not Lower Score:**

The paper is very well presented, and the idea should be notable to a small sub-community within ICLR.

**Metareview: Summary, Strengths And Weaknesses:**

All reviewers agree that this paper, which proposes a structured latent representation for neural PDEs of advective systems, is of high quality and should be accepted.

**Note From Pc:**

if the above contains the word "oral" or "spotlight" please see: "oral" presentation means -> notable-top-5% and "spotlight" means -> notable-top-25%. As stated in our emails, we are disassociating presentation type from AC recommendations